# Velocity Vector Imaging Assessment of Functional Change in the Right Ventricle during Transcatheter Closure of Atrial Septal Defect by Intracardiac Echocardiography

**DOI:** 10.3390/jcm9041132

**Published:** 2020-04-15

**Authors:** Se Yong Jung, Jae Il Shin, Jae Young Choi, Su-Jin Park, Nam Kyun Kim

**Affiliations:** 1Division of Pediatric Cardiology, Congenital Heart Disease Center, Severance Cardiovascular Hospital, Department of Pediatrics, Yonsei University College of Medicine, Seoul 03722, Korea; jung811111@yuhs.ac (S.Y.J.); cjy0122@yuhs.ac (J.Y.C.); 2Department of Pediatrics, Yonsei University College of Medicine, Seoul 03722, Korea; shinji@yuhs.ac; 3Department of Pediatrics, Sejong General Hospital, Bucheon 14754, Korea; 4Department of Pediatrics, Emory University, Atlanta, GA 30322, USA

**Keywords:** right ventricle function, transcatheter closure, atrial septal defect, velocity vector imaging, intracardiac echocardiography

## Abstract

The functional change of the right ventricle (RV) after atrial septal defect (ASD) via transcatheter closure is well known. We assessed the immediate RV functional change using velocity vector imaging (VVI) with intracardiac echocardiography (ICE). Seventy-four patients who underwent transcatheter closure of an ASD were enrolled. VVI in the “home view” of ICE showing the RV was obtained before and after the procedure. Velocity, strain, strain rate (SR), and longitudinal displacement were analyzed from VVI data, and the changes of these parameters before and after the procedure were compared. The velocity of the RV decreased after ASD transcatheter closure (3.97 ± 1.48 to 3.56 ± 1.4, *p* = 0.024), especially in the RV inlet and outlet. The average strain decreased (−19.21 ± 5.79 to −16.87 ± 5.03, *p* = 0.002), as did the average SR (−2.28 ± 0.64 to −2.03 ± 0.61, *p* = 0.006). The average longitudinal displacement did not differ. With the VVI technique, we could clearly observe RV functional change immediately after transcatheter closure of the ASD. RV functional change with regional difference may reflect the heterogeneity of volume reduction and suggest subclinical RV dysfunction. These findings can enhance our understanding of the physiologic changes in the RV during reverse remodeling.

## 1. Introduction 

A secundum atrial septal defect (ASD) is a common congenital heart disease with a left-to-right shunt that induces volume overload in the right ventricle (RV), and may eventually cause RV myocardial deformation. ASD is routinely treated via transcatheter closure under fluoroscopic and echocardiographic guidance with modalities such as intracardiac echography (ICE).

ICE is a multielement, steerable system that yields useful images during intracardiac procedures [1,2,3]. Image guidance with ICE enhances the safety and success of intracardiac procedures [4,5,6,7]. However, ICE is primarily used for the accurate observation of intracardiac anatomical structure. Only a few reports have focused on the functional aspects of ICE. 

Vector velocity imaging (VVI) is an advanced quantification method of echocardiography that can be used for cardiac function assessment by measuring the velocity of the myocardium. VVI has proven clinically useful in many different patients and can be applied in various clinical settings [8,9,10,11].

In this study, we assessed the immediate functional change in the RV of patients undergoing transcatheter closure of a secundum ASD using the VVI method of ICE, which is routinely performed during such procedures. To the best of our knowledge, this is the first such study on the use of this model. 

## 2. Methods

### 2.1. Patients

During two study years, 155 patients underwent transcatheter closure of an ASD at Severance Cardiovascular Hospital, Korea. All the patients were diagnosed with ASD based on the findings of transthoracic echocardiography prior to the procedure and considered suitable candidates for transcatheter device closure. Among these patients, those with other hemodynamically significant intracardiac lesions, moderate-to-severe pulmonary hypertension, or RV dysfunction were excluded. Conclusively, 74 patients who agreed to participate in this study were enrolled.

### 2.2. ASD Device Closure and ICE Imaging Protocol

All the patients underwent diagnostic cardiac catheterization to verify their hemodynamic state. Transcatheter closure of ASD was performed only when the ratio of pulmonary blood flow to systemic flow (Qp/Qs ratio) was >1.5. Device closure was performed under ICE and fluoroscopic guidance. 

ICE imaging was performed with the Sequoia ultrasound system (Siemens Medical Solutions, Inc., Mountain View, CA, USA). The ICE catheter used in the present study was the ACUSON AcuNav™ 8F Ultrasound Catheter (Siemens Medical Solutions USA, Inc.), which has a side-viewing, 64-element, linear-phased array with four-way steering capability. 

The detailed ICE guidance protocol has been described in our previous study [4]. In brief, an 8-Fr sheath was inserted through the left femoral vein in all the patients, after which the ICE catheter was inserted. The ICE catheter was positioned in the right atrium before the procedure, and the ultrasound probe was directed towards the tricuspid valve, which represented the “home view”; thereafter, images for functional assessment were obtained (Figure 1). Another ICE image of the same “home view” was obtained after the procedure, and the two images were analyzed and compared. The methods for ASD device closure and ICE imaging were similar to those used for other ASD device closure cases at our center. However, the postimaging process and analysis differed. 

### 2.3. Image Capture and Analysis

To acquire sufficient information, the images were recorded at 60 frames per second. We captured images of a full cardiac cycle through the correlation of two consecutive QRS complexes on an electrocardiogram.

Image analysis was performed using recorded and separately stored raw data. The software used to analyze the data was Siemens Syngo^®^ US Workplace (Siemens Medical Solutions USA, Inc., Malvern, PA, USA). 

Through the analysis of data obtained on VVI, we obtained the peak systolic velocity, strain, strain rate (SR), and longitudinal displacement values. The “home view” RV image was divided into six different areas: right ventricular inlet, midinlet, apex-inlet, apex-outlet, midoutlet, and right ventricular outlet. The change in these parameters before and after the procedure was compared (Figure 2).

To minimize inter- and intra- observer variation, images were obtained by a specialist interventionist (J.Y.C.), whereas the images were analyzed by another well-trained pediatric cardiologist (N.K.K) via the manual subendocardial border tracing method, a total of three times. The mean values were recorded for analysis in the present study. 

### 2.4. Statistical Analysis

Statistical analysis was performed using SAS version 9.2 (SAS Institute, Cary, NC, USA) software. The D’Agostino–Pearson test was used to confirm a normal distribution among the continuous variables. All normally distributed continuous variables were reported as means and standard deviations, while their non-normally distributed counterparts were expressed as medians with ranges. Categorical variables were expressed as frequencies (percentages). Continuous variables before and after the procedure were compared using a chi-squared test, paired t-test, or Wilcoxon test, as suitable.

A linear mixed model was used for the correction of variables such as age, sex, body surface area (BSA), RV pressure (RVP), and Qp/Qs. All the reported *p*-values were two-sided, and a *p*-value of < 0.05 was considered statistically significant.

### 2.5. Ethics Statement

All the patients provided informed consent prior to the transcatheter closure of the ASD. The study was approved by the Institutional Review Board of the Yonsei University Health System (IRB No: 1-2011-0049).

## 3. Results

### 3.1. Baseline Characteristics of the Enrolled Patients

The demographic and hemodynamic data are presented in Table 1. The median patient age was 21.7 (range, 3–59) years and the mean BSA was 1.1 ± 0.5 m^2^. The mean systolic pulmonary artery pressure of the patients was 36.6 ± 10.8 mmHg and the mean Qp/Qs value was 2.2 ± 0.7. Baseline left ventricular ejection fraction (LVEF) was preserved as 67.0 % ± 5.2 %, and the median heart rate (HR) was 78 (range, 45–160) beats per minute. The procedure was successful in all the patients without any complications, and there was no significant change in LVEF (67.0 % to 66.8 %, *p* = 0.739) or HR (78 to 80 beats per minute, *p* = 0.547) immediately after the procedure. Complete ASD closure was confirmed in all the patients within three months of device closure. During the follow-up assessment, none of the patients exhibited evidence of clinical or hemodynamic RV dysfunction.

### 3.2. Velocity in Each RV Segment on VVI

We found that the velocity decreased from the basal to apical segments of the RV. The average velocity of RV flow decreased after ASD device closure (3.97 ± 1.48 to 3.56 ± 1.40 cm/s, *p* = 0.0248, Table 2). According to segmental analysis, the RV inlet and RV outlet showed a significant decrease in velocity, whereas the midoutlet of the RV showed a decrease in velocity after adjusting for clinical variables using the mixed model. 

### 3.3. Strain in Each RV Segment on VVI

The average strain in the RV decreased after ASD device closure (−19.21 % ± 5.79 % to −16.87 % ± 5.03 %, *p* = 0.0021, Table 3). According to segmental analysis, the strain was lower in the midinlet, midoutlet, and RV outlet portions. 

### 3.4. SR in Each RV Segment on VVI

The average SR was lower after device closure (−2.28 ± 0.64 to −2.46 ± 1.01 1/s, *p* = 0.0064, Table 4), even though segmental analysis did not show any significant difference.

### 3.5. Longitudinal Displacement in Each RV Segment on VVI

We found that the longitudinal displacement decreased from the basal to apical segments of the RV. The average longitudinal displacement did not differ before and after device closure (Table 5). However, segmental analysis indicated a decrease in longitudinal displacement in the RV inlet and outlet.

## 4. Discussion

ICE is considered a useful and safe modality for visualizing the heart during intracardiac procedures for congenital heart disease and radiofrequency ablation [4,12]. Functional studies using ICE have been reported, although many are primarily experimental [13,14] or involve an otherwise structurally normal heart with conditions such as cardiac resynchronization therapy. Even though ICE is used worldwide for imaging guidance during interventions for congenital heart disease, no functional studies involving ICE before and after the interventional correction of structural heart disease have been reported thus far. 

To the best of our knowledge, this is the first study to analyze cardiac function before and after the anatomical structure correction of congenital heart disease with transcatheter interventions involving ICE. 

In this study, we successfully analyzed the changes occurring immediately after the intervention by obtaining simple ICE images (“home view”) before and after the procedure. As the “home view” is routinely imaged during transcatheter closure of ASD, no additional time or effort is required for intervention (except that required for postprocedural analysis). As VVI uses the “border detecting algorithm,” analysis remains feasible, even though the image of the entire myocardial layer is not obtained. A degradation of image quality occurs after the implantation of an ASD device in some cases. However, these degraded images were of sufficient quality for VVI analysis. Therefore, VVI could serve as a useful assessment tool of right ventricular functional change before and after transcatheter closure. 

The data obtained in the present study, which showed that the velocity and longitudinal displacement decreased from the basal to apical segments of the RV, were similar to the findings of the previous study involving VVI in normal children [8], as well as to those of studies evaluating other modalities [15,16].

The main findings of this study are as follows: (1) the velocity of the RV myocardium decreased immediately after transcatheter closure, although the change in velocity at various regions differed; (2) the RV strain decreased, although the change in RV strain at various regions did not differ; (3) the RV SR decreased, irrespective of the regions assessed; and (4) longitudinal displacement decreased in the RV inlet and outlet, although the average value remained unchanged.

In particular, we found that the RV myocardial velocity and strain decreased after transcatheter closure of the ASD. This was consistent with the findings from previous studies involving various modalities [17,18]. These data suggest that RV volume reduces immediately after transcatheter closure of the ASD. However, owing to the lack of reference values for the VVI parameters obtained via ICE, we were unable to confirm the previously reported hypotheses for this condition. One hypothesis reported by Pascotto et al. states that the “supernormal” RV velocity due to volume overload actually decreased to normal physiologic values [19], whereas another hypothesis reported by Eyskens et al. states that the septal velocity decreased to subnormal values [17]. Moreover, the reference parameters obtained via Doppler tissue imaging or VVI with transthoracic echocardiography [8] could not be directly applied to our VVI results obtained using ICE. Nevertheless, the heterogeneity of the RV, either in a structurally normal heart or in cases of congenital heart disease, is well known [20,21]. In the present study, the velocity and strain decreased predominantly in the RV inlet and outlet, and not in the apical portion. This can be partially explained by the inherent complexity of the RV anatomy. The myocardial support structure, as the moderator band which connects the interventricular septum to the anterior papillary muscle and crosses the lower portion of RV chamber, is located asymmetrically in the RV anterior wall and apex. The functional assessment of RV is usually hindered by its complex morphology, meaning VVI would be a valuable tool in assessing RV function; therefore, there is a need to further assess and validate its application in diverse clinical settings. In addition, the average SR in the RV was found to have decreased in this study, which reflects the reduction in volume overload. However, no regional differences in the change in SR were noted. These findings enhanced our understanding of the geometric characteristic of the RV related to the reduction in right heart volume overload.

Longitudinal displacement is a reliable measure of RV systolic function [10]. In the present study, the average longitudinal displacement did not change after transcatheter closure of the ASD, which could be attributed to the normal RV systolic function in our study population. However, a decrease in longitudinal displacement was observed at the RV inlet and outlet, which may reflect the presence of subtle RV dysfunction, as well as a change in preload condition. Further investigation is required, particularly in patients with clinical RV dysfunction. Patients with ASD and pulmonary hypertension are good candidates for assessing RV dysfunction. A recent study indicated that these patients could be successfully treated via medication for pulmonary arterial hypertension and subsequent transcatheter closure of the ASD [22]. We believe that this strategy may be useful and will help evaluate the impact of volume and pressure overload on the RV geometry in cases of ASD with pulmonary hypertension using VVI methods in a future study.

The present study had several limitations. Although the trends in the VVI parameters were consistent with those in previous studies involving VVI in normal children [8], normal VVI values via ICE or reference values from cases of the congenital heart disease group have not yet been obtained. Moreover, this study only involved ASD patients without complications such as pulmonary hypertension. Additionally, HR is an important determinant of RV function. Although HR before and after device closure was not different, a wide range of HRs was observed in this study due to the heterogeneous age group. Further studies to clarify the effect of HR on VVI measurement are also needed. Finally, we were unable to obtain follow-up VVI images via ICE due to the invasive nature of this technique. Therefore, the data obtained cannot be directly applied to all ASD patients, even though the data are consistent with those from previous studies. In the future, we will seek to validate the VVI values obtained via ICE imaging by using other tools such as cardiac magnetic resonance imaging or VVI via transthoracic echocardiography, and shall aim to specifically examine patients in particular categories, such as elderly ASD patients or patients with pulmonary hypertension.

## 5. Conclusions

Both structural and hemodynamic changes occur immediately after ASD device closure. This study showed that the VVI methods could obtain a simple and accurate assessment of RV after ASD device closure. With the VVI technique, we could detect the regional differences of RV morphology and the subtle functional variations related to preload change occurring immediately after the transcatheter closure of the ASD. The RV inlet and outlet exhibited a decrease in velocity and strain, which suggests that the effect of RV volume reduction is marked in those areas. Moreover, RV longitudinal displacement was lower in the same areas, which may indicate the presence of subclinical RV systolic dysfunction. Further research is necessary to examine the relationship between changes in VVI measurement and anatomical characteristics of the RV to enhance our understanding of the physiological changes in RV function in congenital heart diseases.

## Figures and Tables

**Figure 1 jcm-09-01132-f001:**
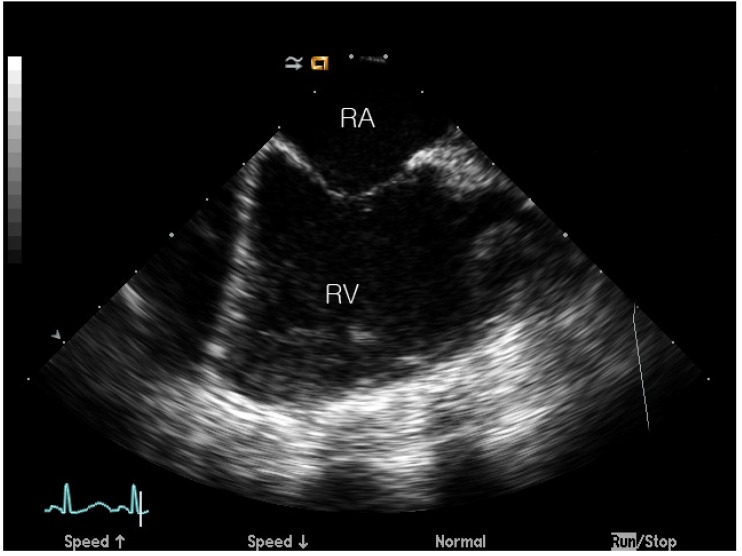
“Home view” of intracardiac echocardiography. RA: right atrium, RV: right ventricle. “Home view” was obtained to analyze the vector velocity image of the right ventricle, before and after the transcatheter closure.

**Figure 2 jcm-09-01132-f002:**
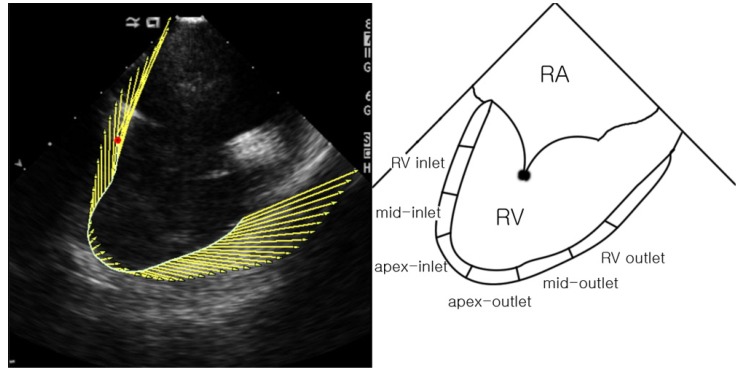
Vector velocity image of the right ventricle in “home view”. RA: right atrium, RV: right ventricle. The right ventricle was divided into six segments, and the velocity, strain, strain rate, and longitudinal displacement of each part was analyzed from the vector velocity image obtained in “home view”.

**Table 1 jcm-09-01132-t001:** Baseline characteristics and hemodynamic data.

Total Patients	N = 74
Age (years)	21.7 (range, 3–59)
Male/Female	20/54
Weight (Kg)	35.9 ± 23.3
Height (cm)	125.9 ± 34.4
BSA (m^2^)	1.1 ± 0.5
Systolic pulmonary artery pressure (mmHg)	37± 11
LVEF (%)	67.0 ± 5.2
Qp/Qs	2.2 ± 0.7
Device Size (mm)	20 (10–36)
Major complication	none
Residual shunt	none

BSA: body surface area, LVEF: left ventricular ejection fraction, Qp/Qs: pulmonary blood flow to systemic blood flow ratios.

**Table 2 jcm-09-01132-t002:** Changes of velocity in each RV segment on VVI (cm/s) after ASD device closure.

Location	Preprocedure	Postprocedure	*p*-Value	Mixed-Model *
RV Inlet	5.05 ± 2.53	4.16 ± 2.34	0.0077	0.01
Midinlet	4.11 ± 2.15	4.03 ± 2.69	0.8358	0.8
Apex-inlet	3.18 ± 2.03	3.33 ± 2.27	0.6264	0.618
Apex-outlet	1.89 ± 1.31	1.70 ± 1.53	0.2402	0.222
Midoutlet	4.12 ± 2.00	3.49 ± 1.63	0.1545	0.04
RV Outlet	5.42 ± 2.09	4.67 ± 1.69	0.0015	0.001
Average	3.97 ± 1.48	3.56 ± 1.40	0.0248	0.022

RV: right ventricle, VVI: vector velocity imaging. *: Adjusted by age, sex, body surface area (BSA), RVP, Qp/Qs.

**Table 3 jcm-09-01132-t003:** Change of strain in each RV segment on VVI (%) after ASD device closure.

Location	Preprocedure	Postprocedure	*p*-Value	Mixed-Model *
RV Inlet	−15.73 ± 10.11	−15.33 ± 12.11	0.7687	0.817
Midinlet	−15.26 ± 9.26	−12.05 ± 9.40	0.015	0.02
Apex-inlet	−16.72 ± 10.84	−14.85 ± 10.21	0.2733	0.22
Apex-outlet	−24.13 ± 9.56	−22.61 ± 11.24	0.3526	0.323
Midoutlet	−25.02 ± 10.41	−21.89 ± 10.59	0.0333	0.028
RV Outlet	−18.42 ± 12.04	−14.51 ± 10.21	0.0312	0.027
Average	−19.21 ± 5.79	−16.87 ± 5.03	0.0021	0.002

ASD: atrial septal defect, RV: right ventricle, VVI: vector velocity imaging. *: Adjusted by age, sex, body surface area (BSA), RVP, Qp/Qs.

**Table 4 jcm-09-01132-t004:** Change of SR in each RV segment on VVI (1/s) after ASD device closure.

Location	Preprocedure	Postprocedure	*p*-Value	Mixed-Model *
RV Inlet	−2.21 ± 1.54	−1.90 ± 1.47	0.1388	0.197
Midinlet	−2.02 ± 1.14	−1.78 ± 1.06	0.0819	0.139
Apex-inlet	−2.07 ± 1.03	−1.93 ± 1.10	0.605	0.458
Apex-outlet	−2.78 ± 1.14	−2.46 ± 1.37	0.0651	0.062
Midoutlet	−2.63 ± 1.31	−2.47 ± 1.01	0.3489	0.338
RV Outlet	−1.96 ± 1.15	−1.79 ± 1.51	0.3187	0.438
Average	−2.28 ± 0.64	−2.03 ± 0.61	0.006	0.008

RV: right ventricle, VVI, vector velocity imaging. *: Adjusted by age, sex, body surface area (BSA), RVP, Qp/Qs.

**Table 5 jcm-09-01132-t005:** Change of longitudinal displacement in each RV segment on VVI (mm) after ASD device closure.

Location	Preprocedure	Postprocedure	*p*-Value	Mixed-Model *
RV Inlet	5.37 ± 3.45	4.22 ± 3.11	0.0045	0.005
Midinlet	4.25 ± 2.81	3.75 ± 2.88	0.221	0.951
Apex-inlet	3.34 ± 2.59	3.25 ± 2.91	0.822	0.816
Apex-outlet	2.77 ± 2.43	2.33 ± 2.13	0.1499	0.155
Midoutlet	5.50 ± 3.19	4.91 ± 2.62	0.1261	0.104
RV Outlet	7.73 ± 3.44	6.87 ± 3.07	0.025	0.02
Average	4.83 ± 2.34	4.22 ± 1.99	0.3161	0.321

ASD: atrial septal defect, RV: right ventricle, VVI: vector velocity imaging. *: Adjusted by age, sex, body surface area (BSA), RVP, Qp/Qs.

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
