# Peer review of "Velocity Vector Imaging Assessment of Functional Change in the Right Ventricle during Transcatheter Closure of Atrial Septal Defect by Intracardiac Echocardiography"

_jcm, 2020, doi:10.3390/jcm9041132_

Round 1

Reviewer 1 Report

Presented study has a relatively small number of patients. Presented study has not brought novelty to literature. The manuscript is well written, however not merit for publication in this journal.

Author Response

We would like to thank the reviewer for reading and reviewing our manuscript and for providing helpful comments. The number of patients seems small, but it is relatively large study considering the volume of device closure of atrial septal defect, and we will perform study involving various clinical situations of atrial septal defect. We hope that you will reconsider the value of this article.

Reviewer 2 Report

Authors present very interesting diagnostic tool to assess hemodynamic changes in right ventricle of patients after atrial septal defect repair. The group of 74 patients, concerning that they had invasive procedure, is quite big to get some conclusions. Despite great results Authors present invasive character of this diagnostic method limits its usefullness, however remains huge improvement in analyzing right ventricular parameters during/after procedure.

If possible please move Figure's 1&2 captions under them, not as they are above.  

Author Response

General comments

Authors present very interesting diagnostic tool to assess hemodynamic changes in right ventricle of patients after atrial septal defect repair. The group of 74 patients, concerning that they had invasive procedure, is quite big to get some conclusions. Despite great results Authors present invasive character of this diagnostic method limits its usefullness, however remains huge improvement in analyzing right ventricular parameters during/after procedure.

We would like to thank the reviewer for reading and reviewing our manuscript and for providing helpful comments. We totally agree, and we will validate the VVI technique and apply to various clinical situations to analyze right ventricular parameters.

Point 1. possible please move Figure's 1&2 captions under them, not as they are above.  

Response 1: Thank you for your comment. We moved the caption of figures appropriately.

Reviewer 3 Report

With VVI technique, the authors observed the RV functional change immediate after transcatheter closure of the ASD. RV functional change with regional difference may reflect the heterogeneity of volume reduction and subclinical RV dysfunction. These findings may enhance our understanding of the physiologic changes in the RV during reverse remodeling.

General comments

This is a manuscript addressing a topic “Velocity Vector Imaging Assessment of Functional Change at the Right Ventricle during Transcatheter Closure of Atrial Septal Defect by Intracardiac Echocardiography”. However, the discussion and conclusions drawn are only partly supported by the results. Some concerns need to be addressed (see comments to the authors for details).

Specific comments

Major

  • In most simple ASD cases, RV dysfunction would not be a clinical problem. Intra-procedure assessment of cardiac function would not have a great impact on clinical practice. Therefore, application for high-risk patients would be informative.
  • Strain and strain rate would be influenced by preload reduction; the results of the present study may have reflected the change of loading condition, but not intrinsic cardiac function.
  • LV function: volume-overload would increase in the LV after ASD closure. LV volume and function may have influence especially on RV inlet motion.
  • Heart rate: as a basic value, R-R interval would be also important. Because the patients’ age ranged very widely in this study, heart rate may be also very different; heart rate is said to influence RV function.

Minor

  • Line 87: The velocity seems that in diastole; are the values in systole or diastole?
  • Line 90: Figure 3 is missing.
  • Lone 103: the statistical methods using a mixed model would be necessary in detail.
  • Line 114 and table 1: the mean value of BSA is lost.
  • Table 3, 4, and 5: the units of values are lost.
  • Table 3, 4, and 5: the title is not complete.
  • Line 152: only the fact that there has been no study would not implicate the importance of the study.
  • Line 173: RV SR did not differ in each segment.
  • Line 193: again, the subtle change in values may not indicate RV dysfunction but change in loading condition.
  • Line 196: the pathology of ASD would be primarily “volume overload” but not pressure overload.
  • Line 221: there was no data of reverse remodeling.

Author Response

Response to Reviewer 3 Comments

General comments

This is a manuscript addressing a topic “Velocity Vector Imaging Assessment of Functional Change at the Right Ventricle during Transcatheter Closure of Atrial Septal Defect by Intracardiac Echocardiography”. However, the discussion and conclusions drawn are only partly supported by the results. Some concerns need to be addressed (see comments to the authors for details).

We would like to thank the reviewer for reading and reviewing our manuscript and for providing helpful comments. The following are our point-by-point responses to the reviewer’s comments.

Specific comments

Major

Point 1

In most simple ASD cases, RV dysfunction would not be a clinical problem. Intra-procedure assessment of cardiac function would not have a great impact on clinical practice. Therefore, application for high-risk patients would be informative.

Response 1: Thank you for your comment. We totally agree, and we already mentioned that application in high-risk patients (elderly patients or pulmonary arterial hypertension patients) is important (line 216-217, 245-246). This exploratory study showed that VVI using ICE is a reliable and important tool for assessing RV function and should be applied in various situations, per your comment.

Point 2

Strain and strain rate would be influenced by preload reduction; the results of the present study may have reflected the change of loading condition, but not intrinsic cardiac function.

Response 2: Thank you for your comment. We totally agree, and already mentioned that the change of strain and strain rate is related to volume reduction. To clarify the meaning, we have added information in the discussion section, which relates volume reduction with possible RV functional changes. Intrinsic cardiac function would be changed as long-term effect after preload reduction, and assessment of long-term cardiac function after ASD device closure using VVI technique would be valuable research.

Point 3

LV function: volume-overload would increase in the LV after ASD closure. LV volume and function may have influence especially on RV inlet motion.

Response 3: Thank you for your comment. We thoroughly understand that LV function influences RV function. It is well known that LV dysfunction could be occurred after ASD device closure due to volume overload for “underloaded” LV especially for elderly patient or patients with large amount of shunt (high Qp/Qs), but this study involved only patients with normal LV function Thus LV function might not affect the RV function in this study. We clarified this in the results section (line 122).

Point 4

Heart rate: as a basic value, R-R interval would be also important. Because the patients’ age ranged very widely in this study, heart rate may be also very different; heart rate is said to influence RV function.

Response 4: Thank you for your comment. This study involved very wide age and heart rate ranges in patients. However, we could not amend the effect of HR on RV function appropriately in the current situation. We added this limitation in the discussion section (lines 237-2400-273).

Minor

Point 5

Line 87: The velocity seems that in diastole; are the values in systole or diastole?

Response 5: The velocity was the peak systolic velocity and others were semi-automatically calculated. We added “peak systolic velocity” to line 88.

Point 6

Line 90: Figure 3 is missing.

Response 6: Thank you for your comment. We apologize for the typing error and changed it to Figure 2 instead of Figure 3 on line 92.

Point 7

Line 103: the statistical methods using a mixed model would be necessary in detail.

Response 7: We used a linear mixed (effect) model to correct for compounding factors, and we added the word “linear” in methods section, line 106.

Point 8

Line 114 and table 1: the mean value of BSA is lost.

Response 8: Thank you for your comment. We added the mean value of BSA in Table 1 and to line 117.

Point 9

Table 3, 4, and 5: the units of values are lost.

Table 3, 4, and 5: the title is not complete.

Response 9: Thank you for your comment. We added the units of values appropriately in the abstract section (line 23, 24 and 25) and results section (Table 2, Table 3, Table 4, and Table 5). We also revised the table titles. We apologize for this error.

Point 10

Line 152: only the fact that there has been no study would not implicate the importance of the study.

Response 10: Thank you for your comment. We just wanted to emphasize the novelty of our study and the possibility for further research.

Point 11

Line 173: RV SR did not differ in each segment.

Response 11: Thank you for your comment. We corrected it to “RV SR at various regions did not differ” on line 190-191, according to our original intention.

Point 12

Line 193: again, the subtle change in values may not indicate RV dysfunction but change in loading condition.

Response 12: Thank you for your comment. We recognized that the phenomenon is primarily related with volume reduction. Nevertheless, a change of RV longitudinal strain is related with a subtle change in RV function as well as volume change.

Point 13

Line 196: the pathology of ASD would be primarily “volume overload” but not pressure overload.

Response 13: Thank you for your comment. We agree completely and changed “pressure overload” to “RV dysfunction” on line 219, and we moved the sentence “Patient with ASD and pulmonary hypertension ~ in the future” to the prior paragraph to correct the logical flow.

Point 14

Line 221: there was no data of reverse remodeling.

Response 14: Thank you for your comment. This study did not involve reverse remodelling. We mean that the VVI technique could enhance our further understanding of the physiological changes including reverse remodeling, once this technique is applied to various clinical situations. We added the word “further” to more accurately clarify our meaning.

Reviewer 4 Report

The authors report a study of the immediate effect of ASD closure upon RV function in 74 patients. Overall this is a very nice paper that is informative and well written.

There are some errors of English language and grammar that would benefit from re-review.

I see real value in reporting the other 81 patients with "other hemodynamically significant intracardiac lesions, moderate-to-severe pulmonary hypertension, or right ventricle dysfunction". I appreciate that you have pitched this paper as a proof of concept paper in the Introduction, but your paper looks more like a physiology paper in a reasonably well-functioning group. The other 81 patients may be more informative. Some of your Discussion (lines 230-6) reinforce my assertion regarding the more c omplex patients.

In the description of the linear mixed model (lines 112-4), you state that you used intracardiac shunt amount. Exactly what do you mean by that? Qp/Qs ?

Similarly, BSA is highly correlated to age and gender in most populations. Please closely examine the model for competing effects of these variables.

Device size is an ordinal variable, not a continuous variable. Please report it as such?

LVEF and sPAP are recorded without decimal places; please report them without decimal places.

When you are reporting change in velocity / strain / strain rate in each RV segment, are you reporting absolute change or percentage change?

Did you measure right heart output or stroke volume in these patients and correlate it with RV functional measures? My question has to do with are these changes merely a reflection of a reduced Qp/Qs, or do they measure some form of true improvement of RV function? You have a section in the Discussion (lines 203-12), but I was unable to discern your conclusions regarding this finding.

Further study is always necessary (lines 269-71), but how do you anticipate your findings informing or improving patient care.

Author Response

General Comments

The authors report a study of the immediate effect of ASD closure upon RV function in 74 patients. Overall this is a very nice paper that is informative and well written.

There are some errors of English language and grammar that would benefit from re-review.

We would like to thank the reviewer for reading and reviewing our manuscript.

Special Comments

Point 1.

I see real value in reporting the other 81 patients with "other hemodynamically significant intracardiac lesions, moderate-to-severe pulmonary hypertension, or right ventricle dysfunction". I appreciate that you have pitched this paper as a proof of concept paper in the Introduction, but your paper looks more like a physiology paper in a reasonably well-functioning group. The other 81 patients may be more informative. Some of your Discussion (lines 230-6) reinforce my assertion regarding the more complex patients.

Response 1: Thank you for your comment. We completely agree, and we think that this VVI method is more valuable to patients with “other hemodynamically significant lesions, moderate-to-severe pulmonary hypertension, or right ventricle dysfunction”. However, we were not sure that this VVI technique was simple and reproducible; thus, we began this exploratory study using patients with relatively stable conditions.

Point 2.

In the description of the linear mixed model (lines 112-4), you state that you used intracardiac shunt amount. Exactly what do you mean by that? Qp/Qs ?

Response 2: Thank you for highlighting this ambiguity. We agree with your suggestion that Qp/Qs is more accurate, and thus we revised to “Qp/Qs” instead of “intracardiac shunt amount” in lines 111-113.

Point 3.

Similarly, BSA is highly correlated to age and gender in most populations. Please closely examine the model for competing effects of these variables.

Response 3: Thank you for your comment. We consulted with statisticians in our university regarding our use of the linear mixed model, and competing or compounding effects were minimized.

Point 4.

Device size is an ordinal variable, not a continuous variable. Please report it as such?

Response 4: Thank you for your comment. Some related articles involving the device size have used mean and standard deviation values despite its ordinal characteristic. Nevertheless, we agree with your suggestion and we have revised our representation of device size as median and range in Table 1.

Point 5.

LVEF and sPAP are recorded without decimal places; please report them without decimal places.

Response 5: Thank you for your comment. We revised the sPAP values to be represented without decimal places as your comment suggests in Table 1. However, the mean LVEF was described with decimal place for comparison of before and after device closure in line 126.

Point 6.

When you are reporting change in velocity / strain / strain rate in each RV segment, are you reporting absolute change or percentage change?

Response 6: Thank you for your comment. All values are reported as absolute change and not percentage.

Point 7.

Did you measure right heart output or stroke volume in these patients and correlate it with RV functional measures? My question has to do with are these changes merely a reflection of a reduced Qp/Qs, or do they measure some form of true improvement of RV function? You have a section in the Discussion (lines 203-12), but I was unable to discern your conclusions regarding this finding.

Point 8.

Further study is always necessary (lines 269-71), but how do you anticipate your findings informing or improving patient care.

Response 7-8: Thank you for your comments. We completely agree that this study cannot solely differentiate that change of VVI value is reflected by volume reduction or by true improvement of RV function. Therefore, as your previous comment (Point 1) suggests, further study involving complex patients (patients with pulmonary hypertension, right ventricular dysfunction, or other complex congenital heart group) is required.

We usually measure cardiac output during catheterization but not before and after the procedure in a general population with ASD. Thus, the correlation between cardiac output and change of VVI could not reflect your major concern.

Indeed, we are investigating the change of VVI value that is reflected by volume reduction or by true RV function in patients with repaired tetralogy of fallot with severe pulmonary insufficiency and RV dysfunction, who require pulmonary valve replacement or pulmonary valve implantation. We usually perform MRI before and after the surgery/intervention for clinical reasons. However, we do not know how to describe this in our present article, which is an investigational study.

Nevertheless, this study shows that the VVI methods could provide simple and accurate assessment of RV function. Therefore, we changed the sentence in line 247-248 to clarify the clinical significance of this study as“ This study showed that the VVI methods could simple and accurate assessment of RV after ASD device closure” instead of “these changes can be successfully quantified by using VVI via ICE”

Round 2

Reviewer 1 Report

The study is well-written and potentially interesting but in my opinion not suitable for publication in this journal

Author Response

(The authors gave the same response as above.)

Reviewer 3 Report

Comments to authors

With VVI technique, the authors observed the RV functional change immediate after transcatheter closure of the ASD. RV functional change with regional difference may reflect the heterogeneity of volume reduction and subclinical RV dysfunction. These findings may enhance our understanding of the physiologic changes in the RV during reverse remodeling.

General comments

This is a manuscript addressing a topic “Velocity Vector Imaging Assessment of Functional Change at the Right Ventricle during Transcatheter Closure of Atrial Septal Defect by Intracardiac Echocardiography”. However, some concerns need to be addressed (see comments to the authors for details).

Specific comments

Major

  • The conclusion could be more specific to the study; Line 248 and 249 are not based on the results; Line 244 and 255: there is no data for “RV reverse remodeling” and these words would not be appropriate at least for conclusion. If the authors want to put them in conclusion, they could describe “Further study is necessary to examine the relationship between the change in velocity etc. and anatomical characteristics of the RV to enhance our understanding of the physiological changes in the RV function after congenital heart diseases.”
  • The authors argued that the different change in velocity and strains can be attributed to the moderator band (but not “moderate band”) (Line 206). More discussion about the possible reasons for the difference in terms of the complex nature of RV anatomy would be informative. The band connects the interventricular septum to the anterior papillary muscle and cross the lower portion of the RV chamber.

Minor

  • Line 24: the values of SR are different between abstract and table: (-2.28 24 ± 0.64 to -2.03 ± 0.61, P=0.006) and (-2.28 ± 0.64 to -2.46 ± 1.01 1/s, P = 0.0064; Table 4) Because the authors mentioned that SR was decreased, the former would be correct.
  • Line 105: the values are also indicated as median (IQ range?). The selection of different description needs any explanation: normal distribution or not?
  • Line 122: 3 months: three months.
  • Table 1: the value of pulmonary pressure: ±is would be necessary;
  • Table 3, 4, and 5: the title is not complete.
  • Line 206: “moderate” would be “moderator”.
  • Line 238: data are not shown for HR change.

Author Response

Response to Reviewer 3 Comments

General Comments

This is a manuscript addressing a topic “Velocity Vector Imaging Assessment of Functional Change at the Right Ventricle during Transcatheter Closure of Atrial Septal Defect by Intracardiac Echocardiography”. However, some concerns need to be addressed (see comments to the authors for details).

We would like to thank the reviewer for reviewing our manuscript and providing helpful comments. The following are our point-by-point responses to the reviewer’s comments.

Special Comments

Major

Point 1

The conclusion could be more specific to the study; Line 248 and 249 are not based on the results; Line 244 and 255: there is no data for “RV reverse remodeling” and these words would not be appropriate at least for conclusion. If the authors want to put them in conclusion, they could describe “Further study is necessary to examine the relationship between the change in velocity etc. and anatomical characteristics of the RV to enhance our understanding of the physiological changes in the RV function after congenital heart diseases.”

Response 1: Thank you for your comment. We completely agree, and have revised the conclusions which could be draw out from our results. Thus, we changed the sentence “we could clearly observed the type of RV functional change” to “could detect the regional differences of RV morphology and subtle functional variations related with preload change ” in line 262-264. We also revised the final sentence as “Further study is necessary to examine the relationship between changes in VVI measurement and anatomical characteristics of the RV to enhance our understanding of the physiological changes in the RV function in congenital heart diseases.” in line 267-271. Indeed, your recommendation is exactly what we intended to communicate in our article.

Point 2

The authors argued that the different change in velocity and strains can be attributed to the moderator band (but not “moderate band”) (Line 206). More discussion about the possible reasons for the difference in terms of the complex nature of RV anatomy would be informative. The band connects the interventricular septum to the anterior papillary muscle and cross the lower portion of the RV chamber.

Thank you for your comment. In clinical settings, the complexity of RV morphology is an obstacle in assessing the RV regional morphological and functional changes, thus we think that VVI is an useful tool to assess RV function since it could differentiate the regional difference of change after ASD device closure. Nonetheless, further assessment and validation is required. We have therefore revised this section of our writing and added a sentence in lines 215-221 as a result of your comments.

 Minor

Point 3

Line 24: the values of SR are different between abstract and table: (-2.28 24 ± 0.64 to -2.03 ± 0.61, P=0.006) and (-2.28 ± 0.64 to -2.46 ± 1.01 1/s, P = 0.0064; Table 4) Because the authors mentioned that SR was decreased, the former would be correct.

Thank you for your comment. We apologize for the typing error and have revised the numbers accordingly in Table 4.

Point 4

Line 105: the values are also indicated as median (IQ range?). The selection of different description needs any explanation: normal distribution or not?

Thank you for your comment. We used median (range) for non-normally distributed continuous variables, and we have revised the section that describes our statistical analyses according to our intentions (in lines 105-110).

Point 5

Line 122: 3 months: three months.

Thank you for your comment. We changed this to three instead of 3 in line 129.

Point 6

Table 1: the value of pulmonary pressure: ± is would be necessary;

Thank you for your comment. It was deleted during a revision and we restored the ± symbol in reference to the pulmonary valve in Table 1.

Point 7

Table 3, 4, and 5: the title is not complete.

Thank you for your comment. We revised the title to begin with “Change of” and end with “after ASD device closure” in Table 3, 4 and 5.

  • Point 8Thank you for your comment. We apologize for the error and have changed it to moderator instead of moderate to align with our original intentions. 
  • Point 9
  •  
  • Line 206: “moderate” would be “moderator”.
  • Line 238: data are not shown for HR change.
  • Thank you for your comment. We added the change of HR in the results section (line 128), and have excluded it from the discussion section for logical sequence.
